# COGNITIONLIGHT: CONTINUE, RETHINK, OR ROLLBACK? SIGNALING FOR PERSONA-AWARE REASONING IN INTELLIGENT AGENTS

## ABSTRACT

In complex, dynamic scenarios, intelligent agents often proceed with overconfidence, repeating errors or switching strategies inconsistently—this leads to hallucinations, particularly in multi-turn or tool-augmented interactions. *Can we equip intelligent agents with human-like cognitive control: to reason adaptively, choose suitable thinking styles, and self-correct in complex, multi-turn tasks?* Inspired by human meta-reasoning, we introduce **CognitionLight** a cognitively inspired control plugin that regulates agent behavior via a symbolic "traffic-light" mechanism. At each reasoning step, CognitionLight computes a multi-dimensional confidence vector and issues one of three symbolic control signals: *Continue* (green), *Switch Persona* (yellow), or *Rollback* (red), dynamically guiding how the agent proceeds. To operationalize the symbolic signals, CognitionLight incorporates a structured **Persona Switching Module**. Upon receiving a control signal, the system selects from five predefined cognitive styles: *Direct, Reflective, Conservative, Tool-Seeking*, and *Contextual*, each implemented via prompt-level behavioral modulation. The choice is guided by a fused representation of task-level uncertainty, feedback consistency, and historical persona performance, enabling adaptive reasoning modulation. Through extensive experiments on multi-turn reasoning benchmarks, we demonstrate that CognitionLight enhances response consistency, reduces hallucinations, and enables dynamic persona adaptation. Our results validate it as a promising framework for integrating human-like meta-reasoning into large-scale agent systems, offering both stability and flexibility in diverse reasoning environments.

## 1 INTRODUCTION

Imagine you are solving a riddle, engaging in a technical dialogue, or drafting an analytical report. At some point, you hesitate—*"Does this make sense?"* You pause, reassess your reasoning, or shift to a different strategy: moving from fast intuition to deliberate analysis, or consulting an external reference. Such moments of self-correction show that human cognition inherently involves regulating the reasoning process: monitoring progress, adapting thinking styles, and revising direction when necessary.

This internal flexibility, often referred to as *meta-reasoning* or *cognitive control*, allows humans to maintain coherence, avoid cascading errors, and remain adaptive across evolving tasks Russell & Wefald (1991); Miller & Cohen (2001) However, as illustrated in Figure 1, most existing AI agents lack such regulatory mechanisms. In single-turn settings, they often produce incorrect outputs with misplaced confidence, without revisiting or correcting their answers. In multi-turn interactions, they frequently lose track of prior context, yielding responses that contradict earlier information. These failures stem not from local errors, but may from the absence of overarching reasoning control—no mechanism to monitor, revise, or adapt. As a result, AI agents exhibit hallucinations, unstable behavior, and limited robustness across a broad spectrum of reasoning scenarios, including multi-step tasks, tool-augmented decision-making, and multi-modal contexts Yao et al. (2022b); Shinn et al. (2023); Mialon & et al. (2023); Zhou & et al. (2023).

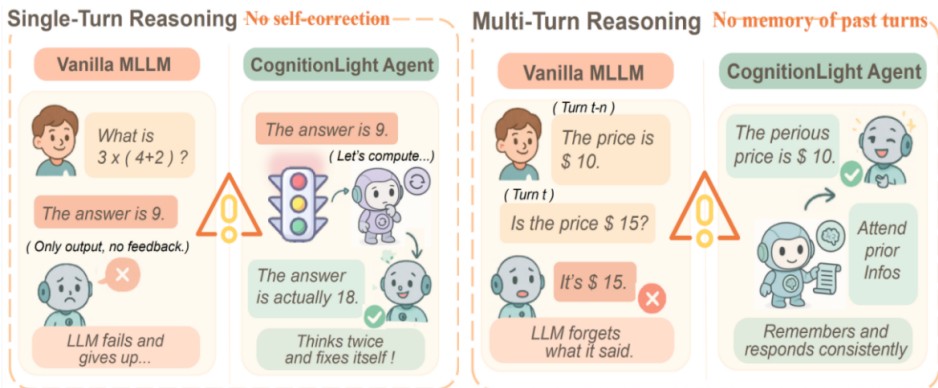

Figure 1: **Illustration of CognitionLight in a single and multi-turn user-agent interaction.** A vanilla agent fails to adjust reasoning style across user turns, giving generic responses (red ×). Our CognitionLight agent dynamically adapts reasoning personas and gets the right answers.

Recent advances have introduced heuristic or feedback-based mechanisms to support step-wise reasoning and error recovery. Chain-of-thought prompting Wei et al. (2022) improves reasoning via intermediate verbalization; ReAct Yao et al. (2022b) integrates tool use with thought processes; Reflexion Shinn et al. (2023) adds self-evaluation and retry loops. RR-MP He et al. (2024) combines reactive and reflective pathways for greater robustness. Toolformer Schick et al. (2023) for failure-aware action revision using tool outcomes and Critic-CoT **?** to frame reasoning as iterative critique and re-decision. Voyager Xu et al. (2023) and CAMEL Liu et al. (2023) develop embodied agents that refine actions over long horizons with minimal supervision. Despite their effectiveness, these approaches lack a structured framework for regulating behavior holistically. Reasoning is not treated as a controlled process where agents can explicitly decide at each step, whether to continue, revise, or adapt based on internal and external feedback. This raises a fundamental question: *Can we equip intelligent agents with human-like cognitive control, enabling them to reason adaptively, select appropriate reasoning styles, and self-correct across complex, multi-turn tasks with evolving demands?*

We address this challenge by decomposing cognitive control into three tightly coupled capabilities: (1) *how to assess the current reasoning status? (First)* (2) *how to adapt the strategy accordingly? (Second)* and (3) *how to regulate the process over time to ensure coherence and flexibility? (Third)* Our proposed framework, **CognitionLight**, unifies these capabilities into a step-wise control architecture, inspired by the metaphor of a traffic signal: green to proceed, yellow to adapt, red to revise.

*First*, to assess reasoning status, CognitionLight introduces a multi-signal sensing mechanism that monitors both internal confidence and external feedback. It computes a confidence vector $\alpha_t$ based on model-internal cues such as uncertainty, multimodal alignment, and tool feedback. This vector is then fused into a scalar confidence score $\gamma_t$, alongside a binary verification signal $\delta_t$. These two signals jointly determine whether the current reasoning path is stable, risky, or failing. *Second*, when uncertainty is detected, the agent adapts its reasoning strategy through **persona switching**. It selects from a predefined set of reasoning styles, including Direct, Reflective, and Tool-Seeking, depending on task demands and signal feedback. This mechanism enables dynamic behavioral modulation while maintaining overall coherence in reasoning. *Third*, a symbolic controller integrates the sensing and adaptation modules to issue step-wise reasoning decisions. A high confidence score leads to a Continue signal; moderate uncertainty prompts a Switch signal; and low confidence combined with failure results in a Rollback decision. These symbolic signals constitute the core control mechanism of CognitionLight, ensuring stable, adaptive, and self-correcting reasoning across complex multi-turn tasks.

CognitionLight offers a general framework for agent reasoning control by aligning three core components: signal computation, persona adaptation, and step-wise decision control. These components are unified under a symbolic cognitive signaling metaphor, which guides the agent's reasoning through interpretable control signals. Through this integration, CognitionLight emulates the flexible

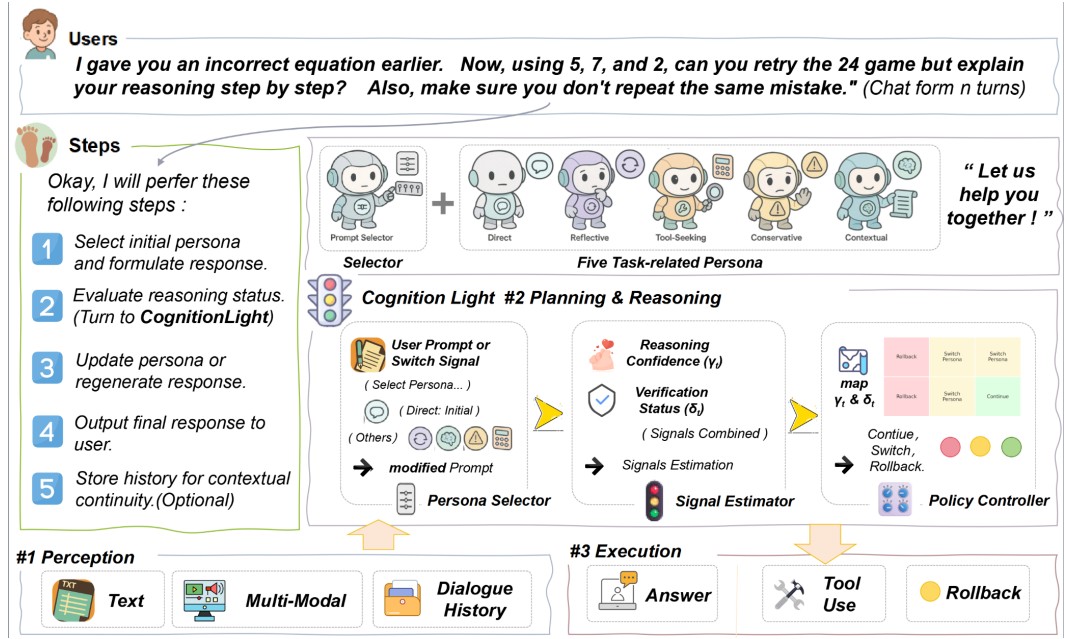

Figure 2: **Overview of the CognitionLight agent workflow for step-by-step multimodal reasoning.** The user provides a multi-turn instruction (top), prompting the agent to initiate a controlled reasoning process. The left panel outlines the core workflow steps. The center highlights the CognitionLight module, consisting of three core components: (1) Persona Selector assigns a persona; (2) Signal Estimator evaluates reasoning status; and (3) Policy Controller issues symbolic decisions. These decisions guide behavior across perception, reasoning, and execution, enabling interpretable, feedback-driven control.

and self-regulatory mechanisms that characterize human intelligent behavior. Our contributions are as follows:

- We propose a **symbolic control mechanism** that translates internal confidence and external feedback into three discrete traffic-light signals: *Continue*, *Switch*, and *Rollback*. These signals provide transparent and fine-grained control over the agent's reasoning process.

- We design a dynamic **persona-switching strategy** that enables agents to choose among five distinct reasoning styles. The strategy adapts personas in real time based on control signals, improving robustness, behavioral flexibility, and error correction in multi-turn tasks.

- We develop a lightweight and general-purpose controller that integrates seamlessly with a wide range of reasoning agents. Experiments on Game24, ALFWorld, and WebShop show consistent improvements in long-horizon task success rates, more effective rollback recovery, and stable persona-switching behavior, all with minimal computational overhead.

## 2 COGNITIONLIGHT: A FRAMEWORK FOR SYMBOLIC COGNITIVE CONTROL

As illustrated in Figure 2, **CognitionLight** serves as a lightweight control layer atop the base agent, interfacing with perception, planning, and execution. It introduces five task-aware personas and a symbolic traffic-light policy to guide step-by-step reasoning. Rather than altering model weights, it supervises the reasoning trajectory—detecting uncertainty, switching strategy, or rolling back when necessary. This supervision is realized through three interdependent modules (Figure 3): the *Signal Estimator* computes behavioral signals from model outputs; the *Persona Selector* adapts the reasoning mode via prompt modulation; and the *Policy Controller* maps internal signals to symbolic decisions. Together, they form an interpretable control loop that enables agents to reason accurately, responsively and reflectively.

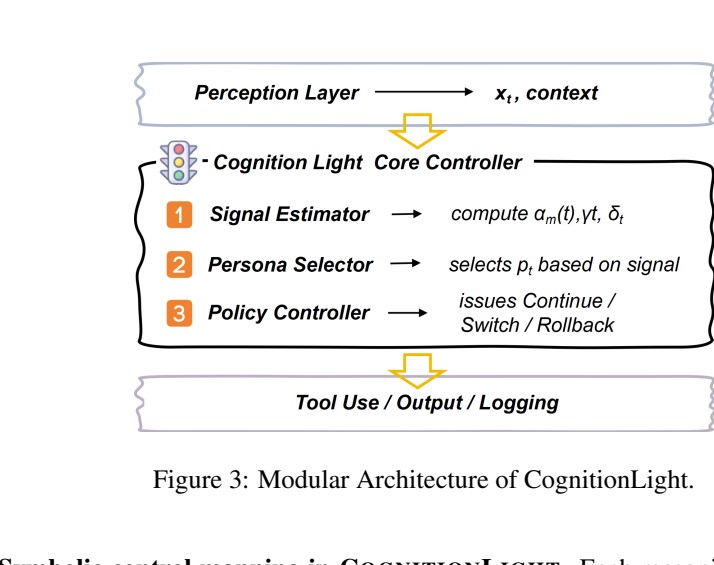

Figure 3: Modular Architecture of CognitionLight.

Table 1: **Symbolic control mapping in CognitionLight.** Each reasoning step is guided by a symbolic signal derived from the scalar confidence $\gamma_t$ and binary correctness flag $\delta_t$. Depending on this combination, the agent executes one of three actions: **Continue**, **Rollback**, or **Switch Persona**. This structured decision space enables dynamic and interpretable adaptation across multi-turn tasks.

| Symbol | Condition $(\gamma_t, \delta_t)$ | Control Decision | Action Type | Persona Behavior |
|--------|------------------|------------------|-------------|------------------|
| Green | $\gamma_t \geq 0.75, \delta_t = 1$ | Continue with current persona | **Continue** | No change (retain current) |
| Red | $\gamma_t \geq 0.75, \delta_t = 0$ | Rollback with current persona | **Rollback** | No change (retry current) |
| Yellow | $0.4 \leq \gamma_t < 0.75, \delta_t = 1$ | **Switch persona** and retry | Soft Switch | Change to alternate style |
| Red | $0.4 \leq \gamma_t < 0.75, \delta_t = 0$ | Rollback and **switch persona** | Full Switch | Abandon current and switch |
| Yellow | $\gamma_t < 0.4, \delta_t = 1$ | **Cautiously switch persona** | Soft Switch | Conservative transition |
| Red | $\gamma_t < 0.4, \delta_t = 0$ | Full rollback and **persona switch** | Full Switch | Reset reasoning with new persona |

## 2.1 SIGNAL ESTIMATOR: HOW TO JUDGE THE REASONING STATUS?

At the core of CognitionLight lies a symbolic control mechanism that enables agents to regulate reasoning progress with structured feedback. Inspired by human meta-cognition, the agent continuously evaluates its own behavior—asking: *Should I proceed, rethink, or rollback?* This is operationalized via three symbolic control signals: green, yellow, and red. Each step $t$ produces two internal judgments: a scalar confidence score $\gamma_t \in [0, 1]$ and a binary correctness flag $\delta_t \in \{0, 1\}$. The former captures uncertainty from internal signals, while the latter reflects outcome-based success (e.g., reaching a goal state or obtaining external verification). Their joint configuration determines the symbolic action, as summarized in Table 1.

**Behavioral Signal Vector** $\vec{\alpha}_t$. To assess the agent's internal reasoning dynamics, CognitionLight computes a five-dimensional behavioral signal vector $\vec{\alpha}_t = [\alpha_t^{(1)}, \ldots, \alpha_t^{(5)}]$, where each component reflects a distinct reasoning property:

- **Confidence Sharpness** ($\alpha_t^{\text{entropy}}$): Measures how peaked or uncertain the model's output distribution is, serving as a proxy for generation confidence.

- **Stability Across Samples** ($\alpha_t^{\text{consistency}}$): Captures whether multiple outputs from the same prompt yield similar responses, indicating deterministic reasoning or behavioral drift.

- **Cross-Modal Alignment** ($\alpha_t^{\text{clip}}$): Reflects the semantic consistency between generated text and visual input, applicable in multimodal scenarios.

- **Tool Interaction Feedback** ($\alpha_t^{\text{tool}}$): Encodes whether external tools (e.g., calculator, API) are invoked successfully, offering task-dependent evidence for correctness.

- **Historical Persona Reliability** ($\alpha_t^{\text{prior}}$): Tracks the recent success rate of the currently active persona, allowing the system to estimate prior reliability in similar contexts.

Table 2: CognitionLight personas with their behavioral intent, prompting strategy, and initialization triggers.

| Persona | Behavior Style | Prompt Guide | Initialization Trigger (Keyword Pattern) |
|---|---|---|---|
| DIRECT | Fast, intuitive response | `Answer directly and concisely.` | Simple prompt, e.g., "translate", "paraphrase" |
| REFLECTIVE | Step-by-step reasoning | `Think step-by-step before answering.` | "explain", "why", "should I…" |
| CONSERVATIVE | Cautious clarification | `Ask clarifying questions if unsure.` | "are you sure", "can you explain" |
| TOOL-SEEKING | Verification via tools | `Use tools if necessary to verify.` | "search", "run Python", "look at chart" |
| CONTEXTUAL | Context-sensitive follow-up | `Use prior turns in this conversation.` | "then", "what about", continuation |

These features are normalized and fused via a learnable aggregation function:

$$\gamma_t = f_\theta(\vec{\alpha}_t) \tag{1}$$

Together, the triplet $(\vec{\alpha}_t, \gamma_t, \delta_t)$ serves as a symbolic diagnostic interface, enabling interpretable and adaptive control at each reasoning step.

## 2.2 PERSONA SELECTOR: HOW TO ADAPT THE REASONING BEHAVIOR?

To robustly support multi-turn reasoning, COGNITIONLIGHT equips the base agent with a set of interpretable reasoning personas and dynamically switches among them based on symbolic signals. Each persona reflects a distinct cognitive strategy, realized via prompt-level control, and is selected or switched in response to evolving confidence and feedback cues.

**Reasoning Personas and Prompt Mapping.** The persona set $\mathcal{P}$ comprises five predefined strategies—DIRECT, REFLECTIVE, CONSERVATIVE, TOOL-SEEKING, and CONTEXTUAL—each paired with a specific behavioral prior and initialization rule. Table 2 summarizes both their behavior styles and trigger patterns used for initial assignment based on the user prompt.

**Signal-Guided Persona Switching.** During reasoning, COGNITIONLIGHT monitors the agent's internal confidence $\gamma_t$ and external feedback $\delta_t$, mapping them into discrete control signals (green/yellow/red). Persona switching is triggered in yellow or red states—when confidence is low or the prior step fails validation. These symbolic signals guide whether to retry reasoning with a different persona. The persona selection policy is formalized as:

$$\pi_{\text{switch}}(t) = \arg\max_{p \in \mathcal{P}} \; \Phi(p \mid \text{persona}_t, \gamma_t, \delta_t, \vec{\alpha}_t) \tag{2}$$

where $\Phi(\cdot)$ scores the utility of switching to persona $p$ based on: (1) Prior effectiveness $\alpha_t^{\text{prior}}$ under similar tasks; (2) Current signals $(\gamma_t, \delta_t)$; (3) Failure mode alignment (e.g., DIRECT→REFLECTIVE).

**Multi-Turn Adaptation.** To avoid repetitive failures, COGNITIONLIGHT tracks each persona's performance over time and updates its prior utility $\alpha^{\text{prior}}$. Personas that repeatedly fail are downweighted, encouraging convergence to more effective reasoning modes. This dynamic adaptation enables the agent to refine its behavior style in long-horizon tasks. Figure 4 visualizes this evolving persona trajectory.

## 2.3 STEPWISE POLICY CONTROLLER: HOW TO MAINTAIN COHERENT REASONING PATHS?

While symbolic signals offer per-step feedback, long-horizon reasoning demands *trajectory-level control*. CognitionLight introduces a structured controller to regulate how signals accumulate, how personas evolve, and how rollback maintains semantic coherence.

**Multi-Turn Control Loop.** At each step $t$, the controller performs:

1. **Signal Computation:** $\vec{\alpha}_t \rightarrow \gamma_t$, with correctness flag $\delta_t$;
2. **Symbol Assignment:** $(\gamma_t, \delta_t) \rightarrow s_t \in \{\text{GREEN}, \text{YELLOW}, \text{RED}\}$;
3. **Policy Decision:** Continue, switch persona, or rollback;

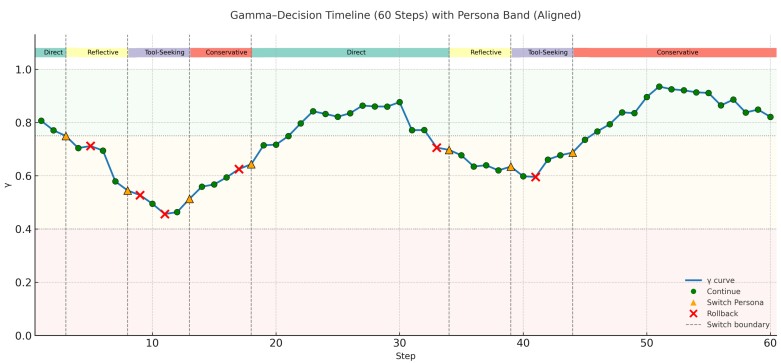

Figure 4: **Gamma-Control Timeline with Persona Switching.** A 60-step reasoning trajectory showing the confidence curve ($\gamma_t$, blue line), control decisions: *Continue* (green dots), *Rollback* (red crosses), and *Switch Persona* (orange triangles), and active personas (top color bar). Background colors denote signal zones: red ($\gamma_t < 0.4$), yellow ($0.4 \leq \gamma_t < 0.75$), and green ($\gamma_t \geq 0.75$). Dashed lines indicate persona switching points. For *Rollback+Switch* cases, control markers are chronologically placed across consecutive steps.

Table 3: **Overall Performance on Three Tasks and LLM Backbones.** Metrics include Game24 Success Rate (G24 SR), ALFWorld Success Rate (ALF SR), WebShop Reward and SR (WS R / WS SR), and averaged success rate across tasks (Avg SR). Best results in **bold**, second-best underlined.

| Method | LLaMA-3.1-8B-Instruct | | | | | GLM4-9B-Chat | | | | | Qwen2.5-14B-Instruct | | | | |
|---|---|---|---|---|---|---|---|---|---|---|---|---|---|---|---|
| | G24 SR | ALF SR | WS R | WS SR | Avg SR | G24 SR | ALF SR | WS R | WS SR | Avg SR | G24 SR | ALF SR | WS R | WS SR | Avg SR |
| Act-only | 0.0 | 36.4 | 27.9 | 58.9 | 21.4 | 0.2 | 75.3 | 23.9 | 66.7 | 33.1 | 3.7 | 76.3 | 25.0 | 42.0 | 35.0 |
| CoT | 0.6 | 34.0 | 28.8 | 51.0 | 21.1 | 0.0 | 73.3 | 36.0 | 60.4 | 38.1 | 4.3 | 83.3 | 26.6 | 57.9 | 38.1 |
| ReAct | 0.5 | 25.9 | 27.2 | 54.3 | 17.9 | 0.5 | 78.9 | 23.8 | 67.2 | 34.4 | 7.7 | 83.0 | 29.9 | 65.2 | 39.2 |
| Reflexion | 7.4 | 47.2 | 33.9 | 64.8 | 29.5 | 8.4 | 77.2 | 36.5 | 68.0 | 40.7 | 15.1 | 86.2 | 42.0 | 72.8 | 47.8 |
| ReAct + Reflexion | 9.2 | 28.8 | 34.1 | 51.9 | 24.0 | 6.5 | 89.4 | 29.2 | 66.5 | 41.7 | 19.3 | 93.5 | 29.9 | 49.0 | 47.6 |
| GA-Rollback | 6.3 | 39.0 | 39.2 | 71.3 | 28.2 | 7.2 | 73.9 | 34.1 | 68.8 | 38.4 | 17.3 | 88.2 | 39.3 | 67.1 | 48.2 |
| C-Light-Rule | 8.1 | 31.6 | 29.3 | 58.7 | 23.0 | 8.9 | 82.5 | 21.4 | 67.8 | 37.7 | 17.2 | 85.8 | 36.8 | 63.3 | 46.6 |
| C-Light-Learn | **9.7** | **49.8** | **42.2** | 70.2 | **33.9** | **11.3** | **88.4** | **44.6** | **71.7** | **48.1** | **22.7** | **95.4** | **47.1** | 70.1 | **55.1** |

4. **Memory Update:** Append $(x_t, a_t, s_t, p_t)$ to $\mathcal{H}$.

The memory buffer $\mathcal{H}$ supports both fine-grained control and global trajectory shaping.

**Cumulative Signal Integration.** Instead of treating $\gamma_t$ in isolation, we compute moving trends:

$$\bar{\gamma}_t = \frac{1}{w} \sum_{i=t-w+1}^{t} \gamma_i, \quad \bar{\delta}_t = \frac{1}{w} \sum_{i=t-w+1}^{t} \delta_i \qquad (3)$$

These trends modulate thresholds and trigger persona re-evaluation when persistent uncertainty arises.

**Semantic Rollback Protocol.** To recover from faulty turns while preserving coherence, rollback follows three constraints:

- **Trusted History:** Retain only steps with $\delta_i = 1$;

- **Persona Compatibility:** Ensure new persona aligns with verified context;

- **Prompt Regeneration:** Resume from last valid $x_i$ with updated persona prompt, guided by $\mathcal{H}_{\leq i}$.

This guarantees correction without disorientation, enabling cognitively traceable multi-turn reasoning.

Table 4: **Ablation Study on CognitionLight Variants (Qwen-2.5-14B-Instruct).** Metrics: Success Rate (SR↑).

| Variant | Game24 SR | ALFWorld SR | WebShop SR | Avg SR |
|---|---|---|---|---|
| Full (C-Light-Learn) | **22.7** | **95.4** | **47.1** | **55.1** |
| Rule (C-Light-Rule, no training) | 17.2 | 85.8 | 36.8 | 46.6 |
| No Rollback | 13.4 | 84.2 | 34.0 | 43.8 |
| No Persona Switching | 8.9 | 78.5 | 26.2 | 37.9 |

Table 5: **Case Study: Step-wise Control Behavior in Game of 24.** At each reasoning step $t$, the agent processes user input and determines a symbolic control signal based on $\gamma_t$ (confidence) and $\delta_t$ (correctness). The signal dictates whether to continue, rollback, or switch persona. The agent transitions across five personas to adapt its reasoning strategy.

| Step $t$ | Input Summary | Persona | $\gamma_t$ | $\delta_t$ | Signal | Action |
|---|---|---|---|---|---|---|
| 1 | Start: "24 Game task" | Direct | 0.92 | 1 | Green | Continue |
| 2 | "Use 3, 8, and 2" | Direct | 0.41 | 0 | Yellow | Switch Persona |
| 3 | "Can you retry another equation?" | Reflective | 0.65 | 0 | Yellow | Switch Persona |
| 4 | "Try evaluating multiple paths" | Tool-Seeking | 0.81 | 1 | Green | Continue |
| 5 | "I still don't see 24" | Tool-Seeking | 0.27 | 0 | Red | Rollback |
| 6 | "Try again, new plan?" | Conservative | 0.69 | 1 | Green | Continue |
| 7 | "Final answer?" | Contextual | 0.87 | 1 | Green | Continue |

## 3 EXPERIMENTS

### 3.1 EXPERIMENTAL SETUP

**Benchmarks and Metrics.** We evaluate CognitionLight on three representative benchmarks that span symbolic reasoning, embodied planning, and tool-augmented decision-making: (1) **Game24**, the classic 24 Game arithmetic reasoning puzzle as included in prior puzzle-reasoning suites Chen et al. (2025), which requires arithmetic reasoning using symbolic expressions; (2) **ALFWorld** Shridhar et al. (2021), a text-based environment for multi-step embodied planning; and (3) **WebShop** Yao et al. (2022a), a product search task involving external tool usage. We report task-specific metrics: binary Success Rate (SR) for Game24 and ALFWorld, and both SR and cumulative Reward for WebShop. All experiments are conducted on three widely-used instruction-tuned models: LLaMA-3.1-8B-Instruct, GLM4-9B-Chat, and Qwen2.5-14B-Instruct.

**Baselines and Control Variants.** We compare CognitionLight with diverse baselines: (1) **Act-only**: direct decoding without reasoning; (2) **CoT** Wei et al. (2022): chain-of-thought prompting; (3) **ReAct** Yao et al. (2023): interleaved reasoning and action; (4) **Reflexion** Shinn et al. (2023): self-reflective verbal reinforcement for failure-aware correction; (5) **ReAct+Reflexion**; and (6) **GA-Rollback** Li et al. (2025): reward-guided rollback strategy. As shown in Table 6, to facilitate flexible control over reasoning behavior, CognitionLight introduces two symbolic controller variants: C-Light-Rule and C-Light-Learn.

**Behavioral Signals and Fusion.** At each reasoning step $t$, CognitionLight computes a behavioral signal vector $\boldsymbol{\alpha}_t = [\alpha_{\text{entropy}}, \alpha_{\text{consistency}}, \alpha_{\text{clip}}, \alpha_{\text{tool}}, \alpha_{\text{prior}}] \in \mathbb{R}^5$, encoding confidence and reliability: (1) $\alpha_{\text{entropy}}$: normalized inverse token entropy; (2) $\alpha_{\text{consistency}}$: exact-match rate across $k = 4$ persona resamples; (3) $\alpha_{\text{clip}}$: normalized CLIP similarity between prompt and image (ViT-B/32); (4) $\alpha_{\text{tool}}$: external tool outcome (1 = success, 0.5 = not used, 0 = failure); (5) $\alpha_{\text{prior}}$: recent success rate of the current persona over the past $m = 20$ steps. These signals are fused into a confidence score $\gamma_t$ that drives symbolic decisions.

**Control Settings and Efficiency.** All runs apply consistent symbolic strategies: (1) five predefined personas with round-robin switching and 2-step lockout; (2) rollback capped at 6 symbolic errors per episode (red lights); and (3) response filtering based on a minimum output confidence threshold

Table 6: **Design Comparison of CognitionLight Controllers.** C-Light-Rule is a zero-shot symbolic controller using rule-based fusion, while C-Light-Learn is a lightweight MLP-based controller trained on labeled trajectories.

| Component | C-Light-Rule (Zero-Shot) | C-Light-Learn (Trained) |
|---|---|---|
| $\alpha$ Signals | $\alpha_{\text{tool}}, \alpha_{\text{prob}}, \alpha_{\text{self}}, 1-\alpha_{\text{entropy}}, \alpha_{\text{hist}}$ | Identical signals |
| Fusion Function ($\gamma$) | $0.30\alpha_{\text{tool}} + 0.25\alpha_{\text{prob}} + 0.15\alpha_{\text{self}} + 0.15(1-\alpha_{\text{entropy}}) + 0.15\alpha_{\text{hist}}$ | $\sigma(W_2 \cdot \text{ReLU}(W_1\boldsymbol{\alpha} + b_1) + b_2)$, 113 params |
| Thresholds | $\tau_{\text{green}}{=}0.75, \tau_{\text{yellow}}{=}0.40$ | $\tau_{\text{green}}{=}$ P90, $\tau_{\text{yellow}}{=}$ P50 (val grid) |
| Actions | $\gamma \geq \tau_{\text{green}}$: Continue; $\tau_{\text{yellow}} \leq \gamma < \tau_{\text{green}}$: Switch; else Rollback | Same three-color logic |
| Persona Policy | Round-robin, switch lock = 2 turns | Same (unchanged) |
| Assistant Filter | Discard if prob-mean ¡ 0.93 | Same |
| Wait-Info Threshold | $k = 6$ (embodied envs) | Same |
| Rollback Cap | $\leq 6$ red lights per episode | Same |
| Training Data | **None** (zero-shot) | $\sim$1k trajectories (success $\leq 3$ steps) |
| Training Cost | Zero (rule-based) | 20 epochs, BCE loss, Adam 1e-3 |
| Deploy Cost | None (static fusion) | One MLP forward ($\sim$5µs/step), LLM frozen |

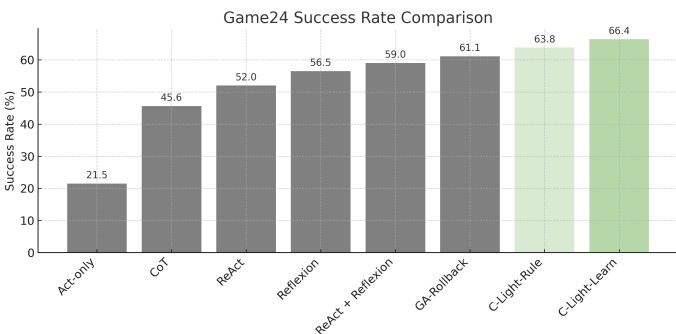

Figure 5: **Game24 Success Rate Comparison.** CognitionLight yields significant improvements over all baselines, particularly in more complex arithmetic compositions.

of 0.93. Notably, all backbone LLMs are kept frozen. Controller decisions are made in real-time, and signal computation adds negligible runtime overhead ($\sim$5µs per step), making CognitionLight deployment highly efficient.

### 3.2 OVERALL PERFORMANCE

To assess the effectiveness of our symbolic control framework, we evaluate eight reasoning paradigms across three representative tasks. Table 3 summarizes the results, measuring Success Rate (SR) and Reward (for WebShop).

**CognitionLight achieves state-of-the-art performance across tasks. C-Light-Learn** outperforms all baselines across 15 metrics, ranking first in 14 of them. On the most challenging setup with Qwen2.5, it achieves 22.7% SR on Game24, compared to 19.3% from ReAct+Reflexion and 17.3% from GA-Rollback. It also leads on ALFWorld with 95.4% SR and on WebShop with a reward of 47.1. Figure 5 illustrates its superior long-horizon decision-making.

**Symbolic control works without training. C-Light-Rule**, our zero-shot variant, outperforms learning-based baselines such as CoT and ReAct in average SR. On LLaMA-3.1, it reaches 8.1% SR on Game24, while ReAct only achieves 0.5%. These results highlight the effectiveness of our interpretable signal-light mechanism.

**Control generalizes across model scales.** CognitionLight provides consistent gains across LLaMA, GLM, and Qwen backbones. The control mechanism operates independently of the model's architecture or scale, confirming its utility as a general symbolic plugin for language agents.

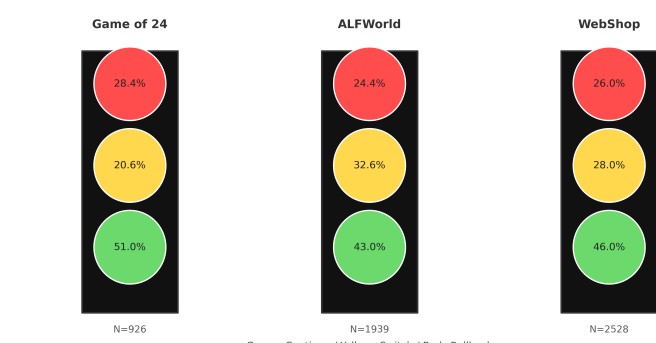

Figure 6: **Rollback vs Switch Outcomes.** Proportions of improved (green), neutral (yellow), and worsened (red) results across tasks. Rollback shows stronger and more stable gains.

## 3.3 MODULE CONTRIBUTION ANALYSIS

To investigate the specific contributions of each symbolic controller within CognitionLight, we conduct an ablation study. Table 4 summarizes the success rates across three benchmark tasks.

**Rollback Enables Self-Correction.** Disabling rollback leads to notable performance drops. In Game24, success rate declines from 22.7% to 13.4%, confirming rollback's role in recovering from errors and stabilizing long-chain reasoning.

**Persona Switching Enables Flexibility.** Without persona switching, performance drops sharply, especially on ALFWorld and WebShop. The average SR falls to 37.9%, showing the importance of dynamically adjusting reasoning style to match task demands.

**Symbolic Control Offers Complementary Gains.** The rule-based controller, though untrained, outperforms single-disabled variants. Rollback aids robustness via error recovery, while persona switching improves adaptability. Their combination yields the best results, with C-Light-Learn further boosting performance through learned signal fusion.

**Rollback and Switching Lead to Positive Outcomes.** Figure 6 shows that most rollback and switching actions result in improved or stable outcomes across all tasks. Improvement is defined by $\Delta$Score $\geq 0.05$ or task success, validating the effectiveness of symbolic interventions.

## 3.4 CASE STUDY: STEP-WISE CONTROL BEHAVIOR

We present a step-wise case from Game of 24 to illustrate CognitionLight's interpretable control process. Table 5 tracks the agent's trajectory, including task input, inferred persona, confidence score $\gamma_t$, correctness $\delta_t$, signal color, and resulting action. At $t=2$, the agent detects low confidence ($\gamma_t=0.41$) and incorrect reasoning ($\delta_t=0$), triggering a yellow signal and persona switch. A second switch occurs at $t=3$, selecting a more reflective mode. The cooldown policy stabilizes behavior, ensuring each persona persists over several steps. At $t=5$, a red signal ($\gamma_t=0.27$) with continued failure prompts a rollback. The agent recovers through conservative planning and succeeds confidently by $t=7$. This trajectory demonstrates how symbolic signals guide adaptive, transparent reasoning.

## 4 CONCLUSION

This paper introduces CognitionLight, a cognitively inspired symbolic control framework for multi-persona reasoning in intelligent agents. By integrating three core components: Persona Selector, Signal Estimator, and Policy Controller, CognitionLight dynamically adapts agent behavior through structured control signals that govern whether to continue, switch, or rollback reasoning steps. Extensive experiments across diverse multi-turn benchmarks demonstrate that our method not only improves task success rates and reasoning stability, but also provides interpretable insight into agent decision-making dynamics. It offers a transparent and generalizable way for controlled reasoning in multimodal agents.

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

## A    ADDITIONAL DISCUSSION OF RELATED WORK

### A.1    FOUNDATIONS OF META-REASONING

Early work on meta-reasoning laid the theoretical foundations for our symbolic control framework. Russell and Wefald formalized core principles for allocating computational resources dynamically during problem solving, demonstrating how an agent can decide when to continue or halt search based on expected utility Russell & Wefald (1991). Miller and Cohen provided an integrative theory of prefrontal cortex function, describing how hierarchical control signals regulate human cognition in a manner analogous to our traffic-light mechanism Miller & Cohen (2001).

### A.2    PROMPT-BASED REASONING AND ACTING

Chain-of-Thought prompting showed that inserting intermediate reasoning steps into prompts significantly improves performance on complex tasks by eliciting latent reasoning in large LMs Wei et al. (2022). ReAct extended this idea by interleaving reasoning traces with external actions, enabling models to query APIs or environments mid-reasoning to reduce hallucination and error propagation Yao et al. (2022b). Reflexion further incorporates episodic self-evaluation loops, letting agents reflect on past failures in natural language and update their future reasoning trajectories without weight updates Shinn et al. (2023).

### A.3    SELF-CRITIQUE AND TOOL LEARNING

Complementary to control signals, recent methods enable models to self-critique and learn tool usage. Toolformer allows LMs to teach themselves when and how to call external APIs in a self-supervised fashion, yielding substantial zero-shot gains on downstream tasks Schick et al. (2023). Critic-CoT introduces a chain-of-thought critic module that generates distant-supervision feedback, emulating System-2 style questioning to systematically refine reasoning outputs Zheng et al. (2024).

### A.4    MULTI-PATH AND REFLECTIVE AGENT ARCHITECTURES

Multi-agent reasoning frameworks such as RR-MP coordinate multiple reactive and reflective agents running in parallel, preventing thought degeneration by aggregating diverse reasoning paths He et al. (2024). This demonstrates the power of structured control loops and collaborative reflection, resonating with our persona-aware, symbolic traffic-light controller.

## B    REPRODUCIBILITY

### B.1    HARDWARE CONFIGURATION

All base models (LLaMA-3.1-8B-Instruct, GLM4-9B-Chat, Qwen2.5-14B-Instruct) remained frozen without any fine-tuning. The CognitionLight controller (C-Light-Learn variant) was trained on a single NVIDIA A100 80 GB GPU for approximately 2 hours using the Adam optimizer with a learning rate of $1 \times 10^{-3}$ over 20 epochs. All inference and evaluation tasks (WebShop, Game24, ALFWorld multi-turn reasoning) were executed in parallel across four A100 GPUs.

### B.2    SOFTWARE AND DEPENDENCIES

The experiments ran on Ubuntu 20.04 LTS with Python 3.10. We relied on transformers 4.39.1, accelerate 0.27.2, openai-clip 1.0, numpy 1.24.1, and torch 2.1.2. Multimodal evaluation used the OpenAI CLIP ViT-B/32 model. Interactive environments (e.g., ALFWorld) were interfaced via the official Gym API and reinforcement-learning agents.

### B.3 Control Strategy Implementation

At each reasoning step, the signal generator emits a five-dimensional action vector $\boldsymbol{\alpha}_t$ comprising confidence entropy, sample consistency, modality alignment, tool feedback, and historical persona success rate. Control lights (red/yellow/green) are determined jointly by $\gamma_t$ (aggregate confidence) and $\delta_t$ (verification success). Persona control offers five preset modes (intuitive, reflective, conservative, tool-oriented, context-based) modulated via prompts. Persona switching employs a two-round lock to prevent oscillation, with a rollback cap of six switches per episode. Responses with confidence below $0.93$ are filtered out.

### B.4 Training and Deployment Efficiency

The rule-based controller (C-Light-Rule) requires no training and thus incurs negligible deployment cost. The learned controller (C-Light-Learn) adds only a lightweight MLP in the inference path, averaging 5 μs per step. Throughout all evaluations, the controller ran stably with minimal impact on overall reasoning time.

## LLM Usage Disclosure

In accordance with the ICLR 2026 policy on large language model (LLM) usage, we disclose that LLM assistance (OpenAI GPT-5 via ChatGPT) was employed during the preparation of this paper. Specifically, the LLM was used for (i) refining the clarity and readability of text, (ii) restructuring sections for better logical flow, and (iii) generating illustrative figure captions and LaTeX formatting templates. All technical content, including problem formulation, theoretical derivations, experimental design, and result interpretation, was conceived, implemented, and validated solely by the authors. The LLM did not contribute to the novelty of the research ideas, data collection, analysis, or conclusions.

