# OpenReview forum: "CognitionLight: Continue, Rethink, or Rollback? Signaling for Persona-Aware Reasoning in Intelligent Agents"
_ICLR.cc/2026/Conference — ICLR 2026 Conference Withdrawn Submission_

### Official Review · Reviewer_XFyY · 2025-10-17

**Soundness:** 2
**Presentation:** 1
**Contribution:** 2
**Rating:** 2
**Confidence:** 4

**Summary:**

This paper introduces CognitionLight, a cognitive control framework designed to address issues where intelligent agents proceed with overconfidence, leading to errors and inconsistencies in complex tasks. Its core is a symbolic control mechanism inspired by a traffic light that, at each reasoning step, issues one of three signals based on internal confidence and external feedback: Continue, Switch Persona, or Rollback. The framework adjusts its reasoning strategy by dynamically switching between five predefined cognitive personas (e.g., Direct, Reflective). Experimental results on benchmarks such as Game24, ALFWorld, and WebShop demonstrate that the method improves task success rates, enhances response consistency, and reduces hallucinations

**Strengths:**

1. Interpretable Control Mechanism: The work proposes an interpretable control mechanism, using a "traffic-light" metaphor to translate complex internal states into clear, actionable decisions (Continue, Switch, Rollback). This provides a transparent way to regulate an agent's reasoning process.

2. Strong Empirical Performance: The framework demonstrates strong empirical performance, outperforming several baseline methods in success rates across the tested benchmarks (Game24, ALFWorld, and WebShop). The results indicate the effectiveness of its structured self-correction approach.

3. Modular and General-Purpose Design: The proposed framework is designed as a modular and lightweight plugin that can be integrated with various agents without altering the base model's weights. Its effectiveness is shown across different LLM backbones, suggesting its potential as a general-purpose controller.

**Weaknesses:**

1.Deficient Presentation and Inadequate Framing of Contribution: The paper suffers from significant presentational and structural flaws that obscure its contributions. The most critical issue is the misplacement of the core experimental results (Table 3) within the methodology section (Chapter 2). The "Experiments" section (Chapter 3) begins on a later page, with its corresponding textual analysis of Table 3 appearing even further on. This disorganization severely disrupts the logical flow. Additionally, the related work discussion in Appendix A is superficial, failing to adequately argue why existing frameworks are insufficient or to precisely situate the novelty of the proposed symbolic control mechanism through a thorough comparison with contemporary architectures.

2.Unverified Foundational Methodological Assumptions: The framework's core mechanism relies on a strong assumption that is not empirically validated within the paper. The concept of "Persona Switching" assumes that a single-line prompt is sufficient to reliably alter an LLM's cognitive style. The paper provides no evidence, either qualitative or quantitative, to demonstrate that the model's outputs genuinely reflect the intended persona shift, making the effectiveness of this key mechanism questionable.

3.Incomplete Evaluation Lacking Critical Efficiency Analysis: The experimental evaluation is insufficient as it omits a crucial aspect of agent performance: cost. The framework's "Rollback" and "Switch Persona" actions inherently increase the number of inference steps. However, the experiments only report success rates and rewards, failing to provide any data on efficiency costs. Furthermore, a critical baseline is missing from the ablation study: an agent that performs "Rollback" but retries with the same persona. This comparison is essential to determine whether the performance gain comes from the intelligent switching policy or merely from having more attempts. Furthermore, there is an omission of relevant and recent literature.

4.Limited Generalizability Due to Reliance on an Idealized Feedback Signal: The framework's control loop is heavily dependent on receiving an immediate, unambiguous, and binary correctness signal (${\delta_t}$) from the environment at each step. This makes the approach well-suited for tasks with clear, discrete success/failure states but severely limits its applicability to more complex, open-ended, real-world problems where such idealized feedback is unavailable.

**Questions:**

1.Regarding Persona Effectiveness: The "Persona Switching" mechanism assumes that single-line prompts can effectively alter an LLM's cognitive style. Have you conducted any qualitative or quantitative analysis (e.g., linguistic feature analysis, performance on cognitive assessment benchmarks) to verify that the model's outputs genuinely reflect the intended personas (e.g., "Reflective" vs. "Direct"), beyond the final task success rate?

2.Regarding Efficiency and Cost Trade-offs: The results in Table 3 show a significant improvement in success rates. However, the rollback and switching mechanisms likely increase the total number of inference steps. Could you provide data on the average number of steps or API calls your method requires to solve a task compared to baselines like ReAct or Reflexion?

3.Regarding the Necessity of Persona Switching: To isolate the benefit of adaptive persona switching from that of simple retries, have you considered an ablation where a "Rollback-only" agent retries a failed step using the same persona? How does such a baseline compare to the full C-Light-Learn model?

4.Regarding Applicability to Tasks with Ambiguous Feedback: The framework's control loop is guided by a binary correctness signal, ${\delta_t}$. How do you envision adapting CognitionLight to tasks where feedback is not immediate, binary, or objective, such as creative writing, text summarization, or open-domain conversation?

5.Regarding the Simplicity of the Switching Strategy: The current implementation uses a fixed, round-robin switching policy with a two-step lockout. Have you explored more adaptive or learnable persona selection policies, for instance, a policy that could learn which persona is the most promising to switch to after a specific type of failure in a given context?

---

### Official Review · Reviewer_aneG · 2025-10-27

**Soundness:** 2
**Presentation:** 3
**Contribution:** 2
**Rating:** 4
**Confidence:** 3

**Summary:**

This work presents CognitionLight, a cognitively inspired control plugin that regulates agent behavior via a symbolic “traffic-light” mechanism. At each reasoning step, CognitionLight computes a multi-dimensional confidence vector and issues one of three symbolic control signals: Continue (green), Switch Persona (yellow), or Rollback (red), dynamically guiding how the agent proceeds. The experiments show that this model improves the LLM agent reasoning performance in three public datasets.

**Strengths:**

1. The paper motivation is clear overall, making it easy to understand this work.

2. The experimental results on three public datasets demonstrate the strengths of this proposed framework.

3. Ablation and case studies provide further insights regarding the effectiveness of the model.

**Weaknesses:**

1. It is unclear about the model design rationales. For example, why does it use five predefined cognitive styles: Direct, Reflective, Conservative, Tool-Seeking, and Contextual? Why not six or four styles?

2. Lack of unique model contribution insights and unclear research gap compared with prior work. From line 72-82, the authors simply listed all prior work with one high-level sentence description and summarized with one simple sentence: "Despite their effectiveness, these approaches lack a structured framework for regulating behavior holistically". What does "structured" exactly mean?

Moreover, figure 1 does not seem to effectively show the unique advantage of this model. It seems that we can simply add a self-correction or memory module to solve the challenges, which are commonly-used approaches in many prior papers.

3. The entire model looks more like a manually crafted complex version of Chain-of-Thought by explicitly defining each step with confidence scores. At first, I thought the confidence score was obtained from model weights or embeddings. However, it turned out to be calculated empirically based on the LLMs' direct outputs. Then this design is similar with many existing self-evaluation of LLMs but with different types of manually crafted scores. Please let me know if I understand the model incorrectly.

Moreover, many features of this model seem to be artificially crafted without rationales. For example, why does the behavioral signal vector have the five scores (Confidence Sharpness to Historical Persona Reliability)? Why not include other possible more score types?

4. Efficiency: With so many modules designed inside, the efficiency and token size of the entire reasoning pipeline can get worse. The overall response time can be much longer than a simple/single-turn reasoning strategy. Moreover, one upgrade of the foundation model (such as GPT-4 from GPT-3.5) may just compensate the improvement gap, without relying on this complex reasoning pipeline.

5. Data pollution/label leakage issue. There are two versions: C-Light-Rule and C-Light-Learn. The C-Light-Rule model without model training shows worse performance in three datasets compared with several baselines. The C-Light-Learn looks better, but it is unclear about how the MLP model was trained. If its training process involved correct trajectories to solve the task, then it may have data pollution/label leakage issues and it is not surprising that it can work better than other models.

6. The related work part is not comprehensive enough, making it hard for the reader to understand the context and detailed research gap. The paper writing and reference issues should be improved, such as '?' symbol in line 77.

**Questions:**

Please check my concerns and questions in Weaknesses section.

---

### Official Review · Reviewer_7Cij · 2025-10-30

**Soundness:** 3
**Presentation:** 3
**Contribution:** 3
**Rating:** 4
**Confidence:** 3

**Summary:**

This paper introduces CognitionLight, a cognitively inspired control framework designed to enhance the reasoning capabilities of intelligent agents, particularly in complex, multi-turn tasks. The core idea is to equip agents with a meta-reasoning mechanism that allows them to dynamically monitor their own state and decide whether to Continue with the current strategy, Switch Persona to adapt their thinking style, or Rollback to a previous correct state upon failure.

The framework operates like a symbolic traffic light. At each reasoning step, a Signal Estimator module computes a multi-dimensional behavioral vector (capturing confidence, consistency, tool feedback, etc.) and fuses it into a scalar confidence score ($\gamma_t$) and a binary correctness flag ($\delta_t$). Based on these signals, a Policy Controller issues one of the three control commands. The Persona Selector module then implements these commands, choosing from five predefined reasoning styles (e.g., Direct, Reflective, Tool-Seeking) to modulate the agent's behavior via prompting.

**Strengths:**

- The core strength of the paper is its cognitively inspired traffic-light framework (Continue, Switch, Rollback). This symbolic control mechanism makes the agent's decision-making process transparent and understandable, which is a significant advantage over more opaque methods.

- The authors have conducted a thorough evaluation across three distinct and challenging benchmarks (Game24, ALFWorld, WebShop), which cover different reasoning modalities. The consistent performance gains across three different LLM backbones demonstrate the generality and robustness of the proposed method. The comparison against a strong suite of baselines, including ReAct and Reflexion, highlights the superiority of the proposed framework.

- The framework is designed in a modular fashion (Signal Estimator, Persona Selector, Policy Controller), making it potentially easy to extend or adapt. For instance, new signals could be added to the estimator, or new personas could be defined. Its ability to work as a lightweight plugin on top of frozen LLMs makes it highly practical and broadly applicable.

- The paper is exceptionally well-written. The concepts are explained clearly, the structure is logical, and the figures and tables are highly effective at conveying the core ideas and results.

**Weaknesses:**

- While the overall framework is novel in its integration, the core ideas it is built upon are well-established. Self-correction is central to frameworks like Reflexion. State rollback has been recently explored in GA-Rollback. Using model confidence (e.g., token entropy) to guide decisions is a common technique. Persona-based prompting is a widely used strategy in prompt engineering. The primary contribution here is the excellent engineering and systematization of these ideas into a unified symbolic controller, rather than a fundamental new concept.

- The framework relies on a fixed set of five predefined personas. The paper does not provide a strong justification for why these specific five were chosen or whether they form a comprehensive set for general-purpose reasoning. This raises questions about the framework's adaptability to tasks that might require a reasoning style not covered by this set. Furthermore, the rule-based controller (C-Light-Rule) uses manually-set coefficients for signal fusion, which appear arbitrary. A more principled explanation for these choices or a sensitivity analysis would be necessary to strengthen this aspect.

- For the C-Light-Learn variant, the paper states it was trained on ~1k trajectories. However, crucial details about the collection and annotation of this data are missing. How was the ground-truth signal labeled for these trajectories? Was it done by humans, or automatically via some heuristics? The performance of the learned model heavily depends on the quality of this data, making these details essential for evaluating the work's soundness.

**Questions:**

- Could you elaborate on the methodology or rationale for selecting the specific five personas? Have you experimented with other personas or considered a mechanism for dynamically generating or adapting personas based on task requirements, rather than relying on a fixed set?

- For the C-Light-Rule variant, what was the process for determining the fusion weights? Are these weights robust across different tasks and models, or do they require tuning? A sensitivity analysis would be very informative.

- Could you please provide more details about the source, collection process, and labeling criteria for the ~1k trajectories used to train the learned controller? Specifically, how were the ground-truth control decisions (Continue, Switch, Rollback) determined for supervision?

---

### Official Review · Reviewer_uCju · 2025-11-01

**Soundness:** 3
**Presentation:** 3
**Contribution:** 2
**Rating:** 4
**Confidence:** 3

**Summary:**

The paper proposes CognitionLight, a cognitively-inspired control plugin that regulates LLM agent behavior through a symbolic "traffic light" mechanism (green: continue, yellow: switch persona, red: rollback). The system incorporates five predefined reasoning personas (Direct, Reflective, Conservative, Tool-Seeking, Contextual) and dynamically switches between them based on confidence signals and feedback.

**Strengths:**

1. The traffic light metaphor provides an interpretable control mechanism that is easy to understand and potentially practical.
2. Evaluation across three diverse benchmarks (Game24, ALFWorld, WebShop) with three different LLM backbones demonstrates breadth.
3. The approach adds minimal computational overhead (~5μs per step) and doesn't require model fine-tuning, making it practically deployable.

**Weaknesses:**

1. The choice of the five signal dimensions (entropy, consistency, CLIP alignment, tool feedback, prior success) lacks theoretical justification from cognitive science
2. The approach assumes a reliable binary correctness signal; this is realistic only for tasks with explicit verification (Game24, ALFWorld, WebShop).
3. The learned controller uses task-specific signals (tool success, CLIP), so calling it “general-purpose” is a bit strong.
4. The paper claims negligible overhead, but some signals (multi-sample consistency, CLIP) are not actually negligible in real multimodal agents.

**Questions:**

1. Why these specific five personas? The selection appears ad-hoc without grounding in psychological or cognitive theories.
Threshold values (0.75, 0.4) seem arbitrary without principled derivation
2. All evaluated environments provide relatively clean success signals (Game24, ALFWorld, WebShop). How would CognitionLight operate in open-ended or ill-posed tasks where δt cannot be obtained or is extremely delayed?
3. C-Light-Learn uses 1k labeled trajectories. How well does this learned controller transfer to unseen backbones (e.g., a larger Qwen or LLaMA variant) without retraining?
4. Persona switching and rollback both consume extra steps. Are the reported gains in Table 3 still present if all methods are constrained to the same total number of LLM calls / tokens?

---

### Note · Authors · 2025-11-12

I have read and agree with the venue's withdrawal policy on behalf of myself and my co-authors.